# The Clinical Significance and Potential of Complex Diagnosis for a Large Scar Area Following Myocardial Infarction

**DOI:** 10.3390/diagnostics15131611

**Published:** 2025-06-25

**Authors:** Valentin Oleynikov, Lyudmila Salyamova, Nikolay Alimov, Natalia Donetskaya, Irina Avdeeva, Elena Averyanova

**Affiliations:** 1Therapy Department, Penza State University, 40, Krasnaya St., 440026 Penza, Russia; l.salyamova@yandex.ru (L.S.); nik.alimovv@gmail.com (N.A.); averyanova-elena90@bk.ru (E.A.); 2Regional Clinical Hospital n/a N.N. Burdenko, 28, Lermontova St., 440026 Penza, Russia; enigmee@rambler.ru

**Keywords:** ST-segment elevation myocardial infarction, cardiac MRI, ischemic injury pattern, multifactorial model

## Abstract

**Background/Objectives**: The aim of this study is to identify markers and develop a multifactorial model for characterizing extensive scar tissue after revascularization in patients with myocardial infarction (MI). **Methods**: A total of 123 patients with MI were examined. The patients underwent contrast-enhanced cardiac magnetic resonance imaging (MRI) with a 1.5 Tesla GE SIGNA Voyager (GE HealthCare, Chicago, IL, USA) on the 7th–10th days from the onset of the disease. At the first stage, we performed a comparative analysis and built a multifactorial model based on the examination results of 92 (75%) patients enrolled from April 2021 to October 2023. These patients formed the group used for model development, or the “modeling group”. The mass of the scar was calculated, including relative to the left ventricular (LV) myocardium mass (M_scar_/LVMM, in %). **Results**: The first subgroup consisted of 36 (39%) patients with a large scar, denoted as “LS” (M_scar_/LVMM > 20%). The second subgroup included 56 (61%) patients with a smaller scar, referred to as “SS” (M_scar_/LVMM ≤ 20%). Logistic regression was used to identify independent factors affecting scar tissue size. A multifactorial model was created. This model predicts M_scar_/LVMM > 20% on MRI. It uses readily available clinical parameters: high-sensitivity troponin I (HscTn I) and N-terminal pro B-type natriuretic peptide (NT-proBNP) levels, and LV relative wall thickness (RWT). We tested the multifactorial model on the “modeling group” (*n* = 31). The sensitivity was 63.6% and the specificity was 85.7%. **Conclusions**: These indicates the feasibility of its application in clinical practice.

## 1. Introduction

Among all cardiovascular diseases, coronary heart disease (CHD) holds one of the top positions in the structure of adult mortality. According to the Global Burden of Disease study, CHD accounted for 9.1 million deaths worldwide in 2019 [1]. Myocardial infarction (MI) is one of the acute forms of CHD. It develops in 3.8% of people younger than 60 and 9.5% of those older than 60 [2]. Sweden recorded 23,200 acute MI cases, resulting in 4700 deaths, with a death rate of 58 per 100,000 population (according to the data of the National Patient Register, the National Cause of Death Register and the National Board of Health and Welfare) [3]. From 1999 to 2019, more than 615,000 people under 65 years of age died from acute MI in the US, equaling 13.4 deaths per 100,000, adjusted for age [4].

The assessment of myocardial viability plays a crucial role in predicting the recovery of left ventricular (LV) contractile function and survival after a heart attack. Modern imaging techniques allow us to identify patients at high risk of cardiac events and optimize therapy to improve their prognosis. Echocardiography (EchoCG) is an affordable technique used to assess the state of the myocardium and global and regional LV systolic function. However, it is not always easy to determine the infarcted zone, as areas of hypokinesia can represent potentially viable stunned or hibernating myocardium [5].

European guidelines for managing patients with STEMI (ST-segment elevation myocardial infarction) and NSTEMI (non-ST-elevation myocardial infarction) recommend the use of cardiac MRI (magnetic resonance imaging) with contrast enhancement as an additional diagnostic tool [6]. This method allows for assessing the structure of the myocardium, revealing the presence and severity of any edema, scarring, heterogeneous zone, and other abnormalities such as microvascular obstruction (MVO) and intramyocardial hemorrhage (IMH) [7,8]. Experts from the American Heart Association emphasize the importance of contrast-enhanced MRI in determining the depth and extent of myocardial damage, especially in identifying subendocardial MI [9]. Studies have shown that larger MRI scars in post-infarction patients are associated with an increased risk of fatal and non-fatal cardiovascular events [10].

However, the high cost of MRI equipment and additional software for the analysis of ischemic and reperfusion injury and the need to involve highly qualified specialists greatly limit the routine use of this method in clinical practice.

### Aim

The aim of this study is to identify markers and develop a multifactorial model for characterizing extensive scar tissue according to cardiac MRI in patients on the 7th–10th day after MI and revascularization.

## 2. Materials and Methods

Between April 2021 and April 2024, 123 patients with STEMI and NSTEMI were examined in the cardiology department with an emergency room and intensive care unit. The Local Ethics Committee approved the protocol and samples of the source documentation, and all patients signed informed consent. The researchers registered the study on the website ClinicalTrials.gov (NCT04347434).

Inclusion criteria: age of 30–70 years; MI, confirmed by a diagnostically significant increase in high-sensitivity I (HscTn I) levels in the blood and changes in the 12-channel electrocardiogram (ECG); and the 1st type of MI under the 4th universal definition [11].

Criteria for non-inclusion: recurrent MI; stenosis of the left main coronary artery trunk greater than 30%; troponin acute heart failure (HF) class III or IV according to Killip at the time of hospitalization or history of chronic HF (CHF) class III or IV; LV hypertrophy greater than 14 mm, as measured by EchoCG; type 1 or 2 diabetes mellitus requiring insulin therapy; stages 4 or 5 chronic kidney disease (CKD); and other severe underlying conditions.

The average age of the patients was 57 (51.5; 61). Among the individuals included in the study, there were 112 men (91%) and 11 women (9%). Pharmacoinvasive revascularization was performed in 46 (37%) cases and 77 (63%) patients had primary percutaneous coronary intervention (PCI). Additionally, 27 (22%) patients had a history of CHD, 93 (76%) suffered from arterial hypertension, and 13 (11%) had diabetes mellitus.

Medication treatment of patients was carried out in accordance with current clinical guidelines [9,10].

We assessed the level of HscTn I in blood serum three times using the Architect i2000 analyzer (Abbott, Abbot Park, IL, USA) during the first two days of hospitalization. Then, we used the maximum value in this work. On days 7–10 after the onset of the disease, patients underwent a series of laboratory and instrumental examinations. The N-terminal pro B-type natriuretic peptide (NT-proBNP) level was determined in blood serum using enzyme immunoassay with the Infinite F50 device (Tecan Austria GmbH, Grödig, Austria).

Using a GE SIGNA Voyager 1.5 T tomograph (GE HealthCare, Chicago, IL, USA) and a single-polar gadolinium-containing contrast agent (gadoteric acid Clariscan™, GE Healthcare, Oslo, Norway), clinicians performed a cardiac MRI. We carried out tissue image analysis using CVI42, version 5.11 (Circle Cardiovascular Imaging Inc., Calgary, AB, Canada). The end-diastolic (EDV) and end-systolic volume (ESV) were determined, followed by the calculation of their indexed value (EDVI and ESVI) and LV ejection fraction (LVEF), LV myocardial mass index (LVMMI) and relative wall thickness (RWT), local contractility index (LCI). The mass of the scar was calculated, including relative to the LV myocardium mass (M_scar_/LVMM, in %). We also evaluated the mass of the peri-infarct zone (PIZ) heterogeneity, its ratio to the mass of the LV (in %), the mass of ischemic damage, including scar tissue and PIZ heterogeneity, and the ratio to the LVMM (in %). Reperfusion injury pattern was analyzed with the help of MVO and IMH (we calculated their mass). The prevalence of myocardial damage was determined by the global contrast index (GCI).

We performed EchoCG on a Vivid E95 ultrasound scanner (GE Healthcare, Chicago, IL, USA). The following LV indicators were recorded: EDV and ESV, with the calculation of indexed values (EDVI and ESVI) and LVEF, LVMMI, and RWT.

For data processing, we used Statistica 13.0 (StatSoft, Tulsa, OK, USA). Parametric data are presented as mean ± standard deviation (M ± SD), while nonparametric data are shown as the median and interquartile range (Me (Q 25%; Q 75%)). We analyzed normally distributed data using Student’s *t*-test for comparisons between groups. The Mann–Whitney test compared values with unequal distributions in two groups. Qualitative indicators were analyzed with the χ^2^ criterion. Logistic regression was used to identify independent factors affecting scar tissue size, and differences were considered statistically significant at *p* < 0.05.

## 3. Results

At the first stage, we performed a comparative analysis and built a multifactorial model based on the examination results of 92 (75%) patients enrolled from April 2021 to October 2023. These patients formed the group used for model development, or the “modeling group”. In the second stage of our research, we assessed the diagnostic value of our multifactorial model using a “validation group” of 31 additional patients (25%). They were examined between November 2023 and April 2024, and their results are presented in Table 1 for comparison with the modeling group.

The MRI results led the researchers to divide the “modeling group” into two subgroups. The first subgroup consisted of 36 (39%) patients with a large scar, denoted as “LS” (M_scar_/LVMM > 20%). The second subgroup included 56 (61%) patients with a smaller scar, referred to as “SS” (M_scar_/LVMM ≤ 20%) [12,13]. Therefore, in the “LS” group, the indicator value was 32.4 (23.4; 39.5) %, and 9.4 (4.5; 15.6) % in the “SS” group (*p* < 0.001). In the 1st subgroup, “pain-to-balloon” time was longer and the number of Q-MI cases on the ECG was greater, as presented in Table 2. Based on other anamnestic and anthropometric data, in terms of medical treatment, the patients in both subgroups were similar.

The analysis of standard LV indices on cardiac MRI demonstrated higher values of EDVI and ESVI in the “LS” subgroup, compared with “SS”, as shown in Table 3. LVEF was higher in patients with M_scar_/LVMM ≤ 20%. In the first subgroup, LVEF ≥ 50% was diagnosed in 17 (47.2%) cases, and in the second subgroup it was in 46 (82.1%) subjects (*p* = 0.004). Moderately reduced LVEF (41–49%) was found in 14 (38.9%) and 9 (16.1%) patients (*p* = 0.014). Low LVEF (≤40%) was seen in 5 (13.9%) and 1 (1.8%) patients, respectively (*p* = 0.022). The level of LCI was 1.5 times higher in patients with a larger amount of scar tissue.

The analysis of volume parameters and LVEF by EchoCG revealed similar results, which are presented in Table 3. The levels of EDVI and ESVI were higher in “LS” patients, while LVEF was higher in the “SS” subgroup.

Although there was no difference in LVMMI between the compared patients, EchoCG detected concentric LV remodeling in seven people (19.4%) from the 1st subgroup and 22 (39.3%) from the 2nd subgroup (*p* = 0.045). We noted eccentric hypertrophy in 10 (27.8%) “LS” patients and six (10.7%) in the “SS” subgroup (*p* = 0.035). There was no significant difference between the subgroups in terms of the frequency of normal LV geometry and concentric hypertrophy.

It is noteworthy that patients with M_scar_/LVMM > 20% had a lower RWT value on EchoCG. This study found pathological values (RWT > 0.42) in 38.9% (14 people) of the first subgroup and 64.3% (36 people) of the second one (*p* = 0.017).

The “LS” subgroup showed higher ischemic damage due to scar tissue and PIZ heterogeneity (*p* < 0.001) compared to the “SS” subgroup. This can be seen in Figure 1. The characteristics of ischemic damage in relation to LV myocardial mass (in %) also were higher in patients with a larger scar area (*p* < 0.001) (Figure 2). We have identified similar differences for GCI. In the 1st subgroup, the indicator was 37.5 (31.6; 49.3), and in the 2nd, 14.7 (8.8; 23.5) (*p* < 0.001). The average number of LV segments affected by infarction in the 1st subgroup was 7.9 ± 2.4, and in the 2nd, 4.5 ± 2.2 (*p* < 0.001).

The results of the reperfusion injury pattern analysis showed that patients with “LS” had unfavorable values for the parameters under study, as shown in Figure 3 and Figure 4. Specifically, in the first subgroup, the frequency of MVO was 3.3 times higher (*p* < 0.001), and IMH 3.7 times (*p* = 0.003), compared with the 2nd one. Additionally, the mass of MVO predominated in individuals with M_scar_/LVMM > 20% (*p* < 0.001) (Figure 3). Analysis of the IMH mass showed no significant differences between these two subgroups (*p* = 0.066) (Figure 4).

Patients with more extensive scarring also had higher levels of cardiac biomarkers. For example, the level of HscTn I in the 1st subgroup was 50,000 (27,590.4; 121,667) pg/mL, while in the second it was 23,267.1 (4960.6; 36,150.4) pg/mL (*p* < 0.001). Similarly, the levels of NT-proBNP were 361.3 (227.7; 609.5) pg/mL and 106.8 (32; 304.5) pg/mL, respectively (*p* < 0.001).

According to the results of a one-factor logistic regression analysis, the following independent variables were identified: “pain-to-balloon” time ≥ 300 min, the presence of a pathological Q wave on the ECG, the duration of CHD, and HscTn I and NT-proBNP; according to EchoCG data, the levels of EDV, ESV, EDVI, ESVI, LVEF, and RWT, including RWT > 0.42, were also identified. These variables demonstrated an effect on scar tissue size, as shown in Table 4. This study aimed to identify readily available indicators of scar tissue size in routine clinical practice. Therefore, MRI characteristics were excluded from the regression analysis.

A one-factor analysis showed no significant correlations between the indicators. Therefore, a multifactorial model was created. This model predicts M_scar_/LVMM > 20% on MRI. It uses readily available clinical parameters: HscTn I and NT-proBNP levels and LV RWT (categorized as 1 if RWT > 0.42, and 0 otherwise). The developed multifactorial model has the following formula:Y = 0.235 + 0.000001X1 + 0.0005X2 − 0.225X3,
where X1 is HscTn I, pg/mL; X2 is NT-proBNP, pg/mL; X3 is 1 for RWT > 0.42 and 0 for RWT ≤ 0.42; and Y is the response variable. If Y is close to 0, this indicates a low probability of achieving the desired outcome. If it is close to 1, this indicates a high probability of it.

In the next stage, we tested the multifactorial model on the “modeling group” (*n* = 31). The diagnostic value of the mathematical formula was determined using a four-fold table (with the calculation of true positive, false positive, true negative, and false negative results). In particular, the sensitivity was 63.6% and the specificity was 85.7%. In turn, the frequency of false positive results was 14.3% and that of false negative results was36.4%. Thus, the value of the response variable Y, close to 1, with high accuracy indicates the presence of M_scar_/LVMM > 20%. However, the presence of a response variable close to 0 does not allow to exclude a large scar size.

## 4. Discussion

The active use of revascularization followed by complex pharmacotherapy has significantly reduced mortality in the acute and subsequent stages of MI. However, the short- and long-term prognosis for patients remains unfavorable [14,15]. Such patients often develop LV remodeling and CHF. According to Jenča D. et al., a history of MI increases the risk of cardiovascular death by four times in patients with HF [16].

Identifying the morphofunctional characteristics of the LV can help identify patients at high risk for fatal and non-fatal complications. EchoCG makes it possible to assess the most important hemodynamic parameters of LV, which is necessary for timely risk stratification [17]. Cardiac MRI with delayed enhancement can identify additional significant characteristics that determine the course of MI: the structure and prevalence of ischemic injury, MVO, and IMH [18,19].

A number of studies have shown a close relationship between the size of the infarcted area relative to the LV mass and recurrent events in the post-infarction period. Heidary S. et al. found that larger MRI-detected scars in MI patients correlated with increased risk of adverse events, including cardiac death, repeated MI or revascularization, worsening of CHF functional class, ventricular tachycardia, ventricular fibrillation or implanted cardioverter defibrillator activation, syncope, and hospitalization [10]. According to the results of 10 randomized trials analysis involving patients with STEMI after primary PCI, the size of MI >17.9% of the LVMM was associated with a high incidence of major adverse cardiac events (MACE) (total mortality, recurrent MI, and hospitalization for HF) [20]. In another study, STEMI and NSTEMI patients with MACE, developed within a year, also showed a larger amount of scar tissue on MRI, amounting to 20.3 (9.0; 28.9) % against 13.1 (5.3; 21.3) % in patients without complications [21]. Therefore, it is extremely important to search for factors characterizing the larger size of the necrosis zone as one of the key predictors of prognosis in the post-infarction period.

We know that increased “pain-to-balloon” time determines the prevalence of MI and the development of the MVO phenomenon, significantly raising the risk of CHF and cardiac death [8,22]. This study showed that patients with larger scar sizes (in %) received PCI later as their primary strategy to restore coronary blood flow and pharmacoinvasive revascularization. As a result, the “LS” subgroup was more often diagnosed with Q-MI.

After acute occlusion of a coronary artery, the ischemic myocardium loses its ability to contract properly, resulting in the development of systolic dysfunction [23]. Cardiac MRI is a reliable method that allows for the accurate assessment of both global and local myocardial contractility [24]. However, in routine practice, EchoCG remains the preferred method for the initial evaluation of LV volume characteristics and LVEF. Their deterioration can be attributed not only to the degree of cardiomyocyte damage but also to the expansion of the infarcted area and the stretching of the scar [25].

In a study by Schwaiger J.P. et al., reduced LVEF on EchoCG and MRI initially characterized patients with an unfavorable prognosis in the long-term postinfarction period, as did a large LV ESV value on EchoCG. A low ejection fraction (LVEF < 52%), measured by both imaging methods, independently predicted MACE [26]. In this current study, LV volume indicators were greater in both MRI and EchoCG of the “LS” patients. LVEF was higher in individuals with a scar size of ≤20%. Interestingly, the RWT, recorded only with EchoCG, showed intergroup differences. Patients with a larger scar area had a significantly higher risk of LCI (1.5 times), suggesting more severe impairment of segmental myocardial contractility.

The ischemic injury zone has a heterogeneous structure and consists of the core of necrosis (scar) and the PIZ heterogeneity. The latter is represented by bundles of surviving cardiomyocytes surrounded by cells that have undergone ischemia and necrosis. This zone has a pronounced arrhythmogenic potential. According to Golcuk E. et al., a size of PIZ heterogeneity > 30% of LVMM is associated with the development of sustained ventricular tachycardia [27]. Ischemic damage mass relative to LVMM was 2.6 times larger in the first subgroup compared to the second, a result of both scar tissue size and PIZ heterogeneity. The close relationship between the size of the scar and the PIZ is probably due to the general blood supply zone of the infarction-related artery, the level of coronary artery damage, and the development of collateral blood flow. An interesting result from comparing ischemic injury zones in the “LS” subgroup showed the scar size to be 2.5 times bigger than the PIZ heterogeneity. “SS” patients showed almost equivalent absolute and relative values. The analysis of GCI and the number of affected segments, which were also significantly higher in “LS” patients, confirmed the differences in the characteristics of the ischemic injury pattern.

The “no-reflow” phenomenon is one of the major complications of revascularization after MI and includes two main phenotypes: MVO and hemorrhagic infiltration of the myocardium. The development of these conditions worsens the prognosis during the post-infarction period [28,29]. The opening of an infarct-related artery within a short period helps to reduce the likelihood of reperfusion injury [30].

Ischemic and reperfusion injury patterns are closely related. Studies have shown this, as demonstrated by Bonfig N. M. et al. Patients with MVO on MRI showed a larger MI size and low LVEF [31]. In the work of Smulders M. W. et al., the size of the infarction, the epicardial surface area of the infarction, and transmurality were predictors of MVO [32]. The value of MI is also associated with IMH. This is associated with a high risk of MACE in patients with STEMI [33,34]. The results of this study confirmed the relationship between reperfusion injury pattern and scar tissue size. In particular, the number of cases of MVO and IMH and the mass of MVO in the “LS” subgroup were 2.1, 3.0, and 3.4 times higher, respectively, compared to the “SS” subgroup. However, they did not differ in terms of IMH weight.

In the analysis of cardiac biomarkers, the concentrations of HscTn I and NT-proBNP were significantly higher in patients with M_scar_/LVMM > 20%. According to Nguyen T. L. et al., HscTn I has a direct correlation with the size of scar tissue and the presence of MVO and an inverse correlation with LVEF [35]. Higher peak values of NT-proBNP 24 h after STEMI were associated with a larger size of myocardial damage and MACE in the long-term period [36].

## 5. Conclusions

On days 7–10 after MI and revascularization patients with a large amount of scar tissue on MRI had unfavorable laboratory, volumetric, and LVEF values, as well as patterns of ischemic and reperfusion injury.

Several risk factors for M_scar_/LVMM > 20% after MI and revascularization were identified using regression analysis. This included the following factors: “pain-to-balloon” time ≥ 300 min, a Q wave on the ECG, the duration of CHD, and HscTn I and NT-proBNP levels. This is also based on echocardiography data (EDVI, ESVI, LVEF, and RWT, including RWT > 0.42). A multifactorial model based on these factors, including HscTn I, NT-proBNP, and RWT in a bimodal distribution, demonstrated good sensitivity and high specificity. This indicates the feasibility of its application in clinical practice.

Limitations of the study. The main limitation of the study is the lack of validation of the proposed model in patients over 70 years, with acute heart failure class III-IV according to Killip at hospitalization and chronic heart failure III-IV functional class in anamnesis.

## Figures and Tables

**Figure 1 diagnostics-15-01611-f001:**
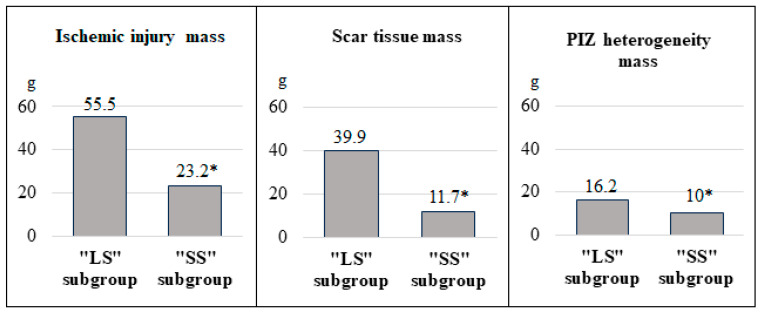
The size of ischemic damage zones (in g) according to MRI data in the compared groups. Note: * *p* < 0.05—significant differences between the groups. “LS” is a large scar and “SS” is a small scar. The graphs show the medians (Me) of the indicators in the subgroups.

**Figure 2 diagnostics-15-01611-f002:**
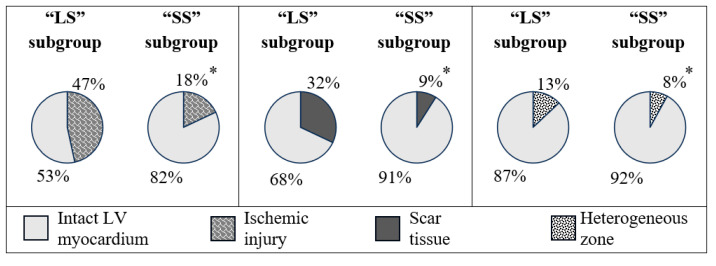
The size of ischemic damage zones relative to the mass of the left ventricular myocardium (in %) according to MRI data in the compared groups. Note: * *p* < 0.05—significant differences between the groups. “LS” is a large scar and “SS” is a small scar. The pie charts show the medians (Me) of the indicators in the subgroups.

**Figure 3 diagnostics-15-01611-f003:**
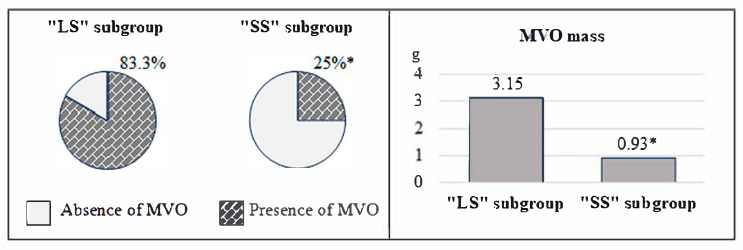
Microvascular obstruction indices according to MRI data in the compared groups. Note: * *p* < 0.05—significant differences between the groups. “LS” is a large scar, “MVO” is a microvascular obstruction, and “SS” is a small scar. The graphs and pie charts show the medians (Me) of the indicators in the subgroups.

**Figure 4 diagnostics-15-01611-f004:**
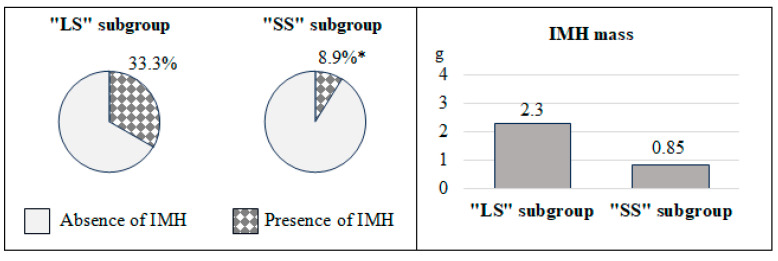
Intramyocardial hemorrhage rates according to MRI data in in the compared groups. Note: * *p* < 0.05—significant differences between the groups. “IMH” is an intramyocardial hemorrhage, “LS” is a large scar, “MVO” is a microvascular obstruction, and “SS” is a small scar. The graphs and pie charts show the medians (Me) of the indicators in the subgroups.

**Table 1 diagnostics-15-01611-t001:** Characteristics of the modeling and validation groups.

Indicators	Modeling Group (*n* = 92)	Validation Group (*n* = 31)	*p*
Age, years	57 (52; 61)	58 (50; 64)	0.392
Male, *n* (%)	85 (92.4)	27 (87.1)	0.371
Female, *n* (%)	7 (7.6)	4 (12.9)	0.371
BMI, kg/m^2^	27.5 ± 3.8	28.1 ± 3.8	0.361
Tobacco smoking, *n* (%)	72 (78.3)	19 (61.3)	0.062
History of CHD, *n* (%)	23 (25)	4 (12.9)	0.159
AH, *n* (%)	83 (90.2)	30 (96.8)	0.245
AH duration, years	8 (8.7)	4 (12.9)	0.496
Pharmacoinvasive revascularization, *n* (%)	31 (33.7)	15 (48.4)	0.144
Primary PCI, *n* (%)	61 (66.3)	16 (51.6)	0.146
“Pain-to-needle” time, min	102.5 (53; 150)	110 (90; 270)	0.341
“Pain-to-balloon” time, min	287.5 (210; 460)	275 (190; 460)	0.422
STEMI, *n* (%)/NSTEMI, *n* (%)	86 (93.5)/6 (6.5)	28 (90)/3 (10)	0.560
Q-MI, *n* (%)/notQ-MI, *n* (%)	53 (57.6)/39 (42.4)	18 (58.1)/13 (41.9)	0.961
Drug therapy			
Lipid-lowering therapy, *n* (%)	92 (100)	31 (100)	1.000
Dual antiplatelet therapy, *n* (%)	92 (100)	31 (100)	1.000
RAAS inhibitors, *n* (%)	88 (95.7)	28 (90.3)	0.261
β-blockers, *n* (%)	82 (89.1)	31 (100)	0.055
Calcium channel blockers, *n* (%)	13 (14.1)	3 (9.7)	0.529
Diuretics, *n* (%)	28 (30.4)	15 (48.4)	0.069

Note: AH—arterial hypertension, CHD—coronary heart disease, MI—myocardial infarction, NSTEMI—non-ST-elevation myocardial infarction, PCI—percutaneous coronary intervention, RAAS—renin-angiotensin-aldosterone system, STEMI—ST-segment elevation myocardial infarction.

**Table 2 diagnostics-15-01611-t002:** Characteristics of patients with different sizes of scar tissue according to contrast-enhanced MRI.

Indicators	“LS” Subgroup (*n* = 36)	“SS” Subgroup (*n* = 56)	*p*
Age, years	58 (52; 61)	56 (52; 60)	0.462
Male, *n* (%)	32 (88.9)	3 (5.4)	0.315
Female, *n* (%)	4 (11.1)	53 (94.6)	0.315
BMI, kg/m^2^	27.6 ± 4	27.4 ± 3.7	0.792
Tobacco smoking, *n* (%)	25 (69.4)	45 (80.4)	0.227
History of CHD, *n* (%)	7 (19.4)	16 (28.6)	0.320
CHD duration, years	1.5 (0.5; 10)	1 (1; 2.5)	0.616
AH, *n* (%)	27 (75)	48 (85.7)	0.197
AH duration, years	5 (3; 9)	5 (2; 9)	0.397
Diabetes mellitus, *n* (%)	3 (8.3)	1 (1.8)	0.136
Pharmacoinvasive revascularization, *n* (%)	12 (33.3)	19 (33.9)	0.953
Primary PCI, *n* (%)	24 (66.7)	37 (66.1)	0.953
“Pain-to-needle” time, min	120 (80; 369.5)	72.5 (44; 120)	0.150
“Pain-to-balloon” time, min	360 (210; 625)	242.5 (150; 380)	**0.009**
STEMI, *n* (%)/NSTEMI, *n* (%)	35 (97)/1 (3)	51 (91)/5 (9)	0.244
Q-MI, *n* (%)/notQ-MI, *n* (%)	31 (86.1)/ 5 (13.9)	22 (39.3)/ 34 (60.7)	**<0.001**
SBP, mm Hg	123 (116; 130)	121.5 (115; 130.5)	0.737
DBP, mm Hg	80 (71; 81)	80 (74.5; 85)	0.266
HR, bpm	71 ± 9.4	69 ± 10	0.336
Drug therapy			
Lipid-lowering therapy, *n* (%)	36 (100)	56 (100)	1.000
Dual antiplatelet therapy, *n* (%)	36 (100)	56 (100)	1.000
RAAS inhibitors, *n* (%)	34 (94.4)	54 (96.4)	0.647
β-blockers, *n* (%)	33 (91.7)	49 (87.5)	0.527
Calcium channel blockers, *n* (%)	5 (13.9)	8 (14.3)	0.957
Diuretics, *n* (%)	14 (38.9)	14 (25)	0.157

Note: AH—arterial hypertension, CHD—coronary heart disease, “LS”—large scar, MI—myocardial infarction, NSTEMI—non-ST-elevation myocardial infarction, PCI—percutaneous coronary intervention, RAAS—renin-angiotensin-aldosterone system, “SS”—small scar, STEMI—ST-segment elevation myocardial infarction.

**Table 3 diagnostics-15-01611-t003:** LV indices according to imaging techniques in the compared groups.

Indicators	“LS” Subgroup (*n* = 36)	“SS” Subgroup (*n* = 56)	*p*
Magnetic resonance imaging			
EDVI, mL/m^2^	84 (72.2; 97.3)	73.8 (65.7; 85.5)	**0.016**
ESVI, mL/m^2^	40.7 (35.6; 48.9)	33.5 (28.4; 38.3)	**<0.001**
LVEF, %	49.1 ± 7.6	55.6 ± 5	**<0.001**
LVMMI, g/m^2^	60.8 (54.9; 72.6)	58 (52.9; 64.2)	0.175
RWT	0.39 (0.34; 0.46)	0.43 (0.38; 0.54)	0.076
LCI	1.95 (1.58; 2.3)	1.31 (1.1; 1.5)	**<0.001**
Echocardiography			
EDVI, mL/m^2^	62.4 (52.8; 70.5)	56.6 (44.9; 63.9)	**0.037**
ESVI, mL/m^2^	28.7 (22.6; 37.2)	23.4 (18.7; 33.4)	**0.005**
LVEF, %	50.9 ± 10.1	55.4 ± 8.8	**0.027**
LVMMI, g/m^2^	113.5 (95.6; 145.5)	107 (92.1; 124)	0.216
RWT	0.39 (0.34; 0.49)	0.48 (0.39; 0.55)	**0.003**

Note: EDVI—the index of the end-diastolic volume, ESVI—the index of the end-systolic volume, LCI—local contractility index, “LS”—large scar, LVEF—left-ventricular ejection fraction, LVMMI—left-ventricular myocardial mass index, RTW—relative wall thickness, “SS”—small scar.

**Table 4 diagnostics-15-01611-t004:** Markers characterizing the presence of scar tissue mass relative to LV myocardial mass > 20%, according to regression analysis.

Indicators	β	SE	B	*p*
One-factor analysis				
“Pain-to-balloon” time ≥ 300 min	0.252	0.111	0.245	0.026
Q-MI	0.462	0.093	0.457	<0.001
CHD duration	0.431	0.197	0.065	0.040
HscTn I, pg/mL	0.477	0.093	0.000	<0.001
NT-proBNP, pg/mL	0.418	0.104	0.001	<0.001
EDVI, mL/m^2^	0.245	0.102	0.008	0.019
ESVI, mL/m^2^	0.299	0.101	0.015	0.004
LVEF, %	−0.231	0.103	−0.012	0.027
RWT	−0.287	0.101	−1.118	0.006
RWT > 0.42	−0.249	0.102	−0.244	0.017
Multifactorial analysis				
Free term	–	–	0.235	0.011
HscTn I, pg/mL	0.353	0.096	<0.001	<0.001
NT-proBNP, pg/mL	0.337	0.096	<0.001	0.001
RWT > 0.42	−0.226	0.095	−0.225	0.020

Note: β—the regression coefficient, B—the angular coefficient indicating the average feature change for each unit of variable increase, CHD—coronary heart disease, EDVI—the index of the end-diastolic volume, ESVI—the index of the end-systolic volume, HscTn—high-sensitivity troponin, LVEF—left-ventricular ejection fraction, MI—myocardial infarction, NT-proBNP—N-terminal pro b-type natriuretic peptide, *p*—reliability, RWT—relative wall thickness, SE—standard error.

## Data Availability

The original contributions presented in this study are included in the article. Further inquiries can be directed to the corresponding author.

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
