# Peer review of "The Clinical Significance and Potential of Complex Diagnosis for a Large Scar Area Following Myocardial Infarction"

_diagnostics, 2025, doi:10.3390/diagnostics15131611_

Round 1
Reviewer 1 Report
Comments and Suggestions for Authors
Thank you very much for sending the review of the interesting work: The Clinical Significance and Potential of Complex Diagnosis for a Large Scar Area Following Myocardial Infarction.
The manuscript is interesting
Minors:
1. Please include a graphic abstract with highlighted results. This will significantly enhance the
educational value of the work.
2. Please expand the subsection Aims and place it separately
3. Please simplify Figure 1 and Figure 2. They are hard to read.
4. Please include the subsection: Limitations of the Study
Author Response
Response to Reviewer 1 Comments
Thank you very much for taking the time to review this manuscript. Please find the detailed responses below and the corresponding revisions/corrections highlighted/in track changes in the re-submitted files.
Point-by-point response to Comments and Suggestions for Authors
- Please include a graphic abstract with highlighted results. This will significantly enhance the educational value of the work.
Done
- Please expand the subsection Aims and place it separately
Done
- Please simplify Figure 1 and Figure 2. They are hard to read.
Done
- Please include the subsection: Limitations of the Study
Done
- The English could be improved to more clearly express the research.
Corrected
Thank you very much for your comments!
Reviewer 2 Report
Comments and Suggestions for Authors
The manuscript presents a well-structured clinical study evaluating the predictors and consequences of large myocardial scar formation in patients following myocardial infarction using both cardiac MRI and echocardiographic parameters. The authors also propose a multifactorial model based on routinely available biomarkers to identify patients at risk for extensive scarring.
Consider discussing the clinical implications of false negatives and false positives in model predictions, especially given the moderate sensitivity (63.6%).
A few grammatical errors and stylistic inconsistencies (e.g., "prevailed" vs. "was more frequent") are present and can be polished.
Please mention the potential role of these parameters while scoring patients for primary prevention with ICD therapies. Please mention this issue citing 'Comparison of mortality prediction scores in elderly patients with ICD for heart failure with reduced ejection fraction' and 'Prognostic nutritional index as the predictor of long-term mortality among HFrEF patients with ICD.
The manuscript provides a useful contribution to risk stratification in post-MI patients. The multifactorial model could be valuable in clinical settings, especially where MRI is not readily available.
Author Response
Response to Reviewer 2 Comments
Thank you very much for taking the time to review this manuscript. Please find the detailed responses below and the corresponding revisions/corrections highlighted/in track changes in the re-submitted files.
Point-by-point response to Comments and Suggestions for Authors
Consider discussing the clinical implications of false negatives and false positives in model predictions, especially given the moderate sensitivity (63.6%).
Done
A few grammatical errors and stylistic inconsistencies (e.g., "prevailed" vs. "was more frequent") are present and can be polished.
Done
Please mention the potential role of these parameters while scoring patients for primary prevention with ICD therapies. Please mention this issue citing 'Comparison of mortality prediction scores in elderly patients with ICD for heart failure with reduced ejection fraction' and 'Prognostic nutritional index as the predictor of long-term mortality among HFrEF patients with ICD.
Done (#15)
Thank you very much for your comments!
Reviewer 3 Report
Comments and Suggestions for Authors
The manuscript addresses a critical area of myocardial infarction (MI) and revascularisation. The use of multimodal imaging is essential and fundamental in the context of MI. Here are some suggestions for improvement:
- The abstract is well-written.
- The introduction is informative, but would benefit from a more concise presentation of the research gap and hypothesis.
- The results, tables, and figures are clear and well-presented.
- It would be helpful to indicate whether echocardiographic and CMR assessments were conducted by one or more blinded operators to minimise bias, and to specify inter- or intra-operator variability.
- Consider discussing potential pathophysiological mechanisms that link the size of a scar in MI to the presence of PIZ.
- Think about including a graphical abstract illustration.
Author Response
Response to Reviewer 3 Comments
Thank you very much for taking the time to review this manuscript. Please find the detailed responses below and the corresponding revisions/corrections highlighted/in track changes in the re-submitted files.
Point-by-point response to Comments and Suggestions for Authors
- The abstract is well-written.
Thank you very much
- The introduction is informative, but would benefit from a more concise presentation of the research gap and hypothesis.
Done
- The results, tables, and figures are clear and well-presented.
Thank you very much
- It would be helpful to indicate whether echocardiographic and CMR assessments were conducted by one or more blinded operators to minimise bias, and to specify inter- or intra-operator variability.
The aim of the study was not to evaluate inter- or intra-operator variability.
- Consider discussing potential pathophysiological mechanisms that link the size of a scar in MI to the presence of PIZ.
Done
- Think about including a graphical abstract illustration.
Done
Thank you very much for your comments!